# Coexistence Scheme for Uncoordinated LTE and WiFi Networks Using Experience Replay Based Q-Learning

**DOI:** 10.3390/s21216977

**Published:** 2021-10-21

**Authors:** Merkebu Girmay, Vasilis Maglogiannis, Dries Naudts, Adnan Shahid, Ingrid Moerman

**Affiliations:** IDLab, Department of Information Technology, IMEC, Ghent University, Technologiepark Zwijnaarde 15, B-9052 Ghent, Belgium; vasilis.maglogiannis@ugent.be (V.M.); dries.naudts@ugent.be (D.N.); adnan.shahid@ugent.be (A.S.); ingrid.moerman@ugent.be (I.M.)

**Keywords:** LTE-U, IEEE802.11, experience replay, Q-learning, coexistence

## Abstract

Nowadays, broadband applications that use the licensed spectrum of the cellular network are growing fast. For this reason, Long-Term Evolution-Unlicensed (LTE-U) technology is expected to offload its traffic to the unlicensed spectrum. However, LTE-U transmissions have to coexist with the existing WiFi networks. Most existing coexistence schemes consider coordinated LTE-U and WiFi networks where there is a central coordinator that communicates traffic demand of the co-located networks. However, such a method of WiFi traffic estimation raises the complexity, traffic overhead, and reaction time of the coexistence schemes. In this article, we propose Experience Replay (ER) and Reward selective Experience Replay (RER) based Q-learning techniques as a solution for the coexistence of uncoordinated LTE-U and WiFi networks. In the proposed schemes, the LTE-U deploys a WiFi saturation sensing model to estimate the traffic demand of co-located WiFi networks. We also made a performance comparison between the proposed schemes and other rule-based and Q-learning based coexistence schemes implemented in non-coordinated LTE-U and WiFi networks. The simulation results show that the RER Q-learning scheme converges faster than the ER Q-learning scheme. The RER Q-learning scheme also gives 19.1% and 5.2% enhancement in aggregated throughput and 16.4% and 10.9% enhancement in fairness performance as compared to the rule-based and Q-learning coexistence schemes, respectively.

## 1. Introduction

Recently, the modern industry has expanded the deployment of wireless networks in search of effective networking solutions that can improve network performance. The rapid increase of wireless network deployments in the industry, along with the quick penetration of wireless networks consumer devices such as smartphones and tablets, has resulted in an exponential increase in wireless traffic demand. Furthermore, the Internet-of-Things (IoT), which connect an unprecedented number of intelligent objects to next-generation mobile networks, consume a major chunk of the wireless spectrum [1].

As the wireless network industry keeps expanding, the licensed spectrum has become a scarce resource and the wireless network industry has shifted its focus towards exploiting unlicensed spectrum as an efficient approach of addressing spectrum shortages and customers’ fast-expanding need for data traffic [2]. In this regard, many fourth-generation Long Term Evolution (LTE) versions, such as LTE unlicensed (LTE-U), LTE licensed assisted access (LAA), and MulteFire [3], have been proposed to use the unlicensed 5 GHz band, which is mostly used by the WiFi network. LTE-U is a technology developed as the first standard for unlicensed spectrum sharing and it has been proposed by the LTE-U forum in 3GPP release 12 [4] of the LTE specifications. The LTE-U technology extends the LTE operation to the unlicensed spectrum; that is, the LTE standard protocol can be used to communicate on the unlicensed band, and the carrier aggregation technology can aggregate licensed and unlicensed bands. Part of the data transmission in the licensed spectrum is thus shifted to the unlicensed band [3].

When LTE is introduced into a common unlicensed spectrum, it will certainly cause issues while competing and coexisting with other unlicensed communications technologies that use the same spectrum. In traditional communication technologies that use unlicensed spectrum for data transmission, such as WiFi, channel access can only be obtained competitively to achieve fair spectrum sharing. On the other hand, LTE technology, which is initially developed in the licensed spectrum, needs complete spectrum control during its data transmission [4].

LTE eNodeB (eNB) performs centralized scheduling of wireless resources to maximize spectral efficiency. Even in the absence of data traffic, LTE uses continuous signal transmission with minimum time gaps within the allocated resources. On the other hand, WiFi uses the carrier sense multiple access with collision avoidance (CSMA/CA) protocol to coexist with other unlicensed band wireless technologies. If the unlicensed spectrum is used as new LTE frequency bands, the transmission of LTE-U will cause significant interference to WiFi. As a result, it is important to design a reasonable and equitable coexistence scheme to ensure fair and efficient coexistence between the two technologies [2].

Carrier Sense Adaptive Transmission (CSAT) is one of the LTE/WiFi coexistence mechanisms which have been proposed by Qualcomm [5]. In CSAT, LTE uses ON and OFF duty-cycle periods to give transmission opportunities (TXOP) to co-located WiFi networks. During an OFF period (mute period), LTE does not transmit and this gives the opportunity to other WiFi networks operating in the same spectrum to transmit. On the other hand, LTE will access the channel during an ON period. The duration of the LTE ON and OFF periods are defined by the eNB to achieve higher aggregated throughput while maintaining fairness between the technologies. However, there is no standard algorithm used to select the optimal ON-OFF time ratio, and the selection of this optimal ON–OFF duration ratio is open for research. LTE Licensed Assisted Access (LAA), on the other hand, is another coexistence approach that employs a mechanism standardized by 3GPP known as Listen Before Talk (LBT). Before any transmission in unlicensed spectrum, LBT performs Clear Channel Assessment (CCA) to determine transmission opportunities using energy detection [6]. As it does not follow the same regulations as WiFi, CSAT is widely seen as more aggressive and less fair than LBT [7]. However, if CSAT is properly designed, it can provide the same level of fairness as compared to LBT. For this reason, we propose an efficient and fair CSAT-based LTE-U and WiFi coexistence scheme in this article.

In our previous work [8], a Convolutional Neural Network (CNN) is proposed that senses WiFi saturation. This work uses this WiFi saturation sensing model to develop an LTE-U/WiFi coexistence mechanism that selects optimal LTE ON and OFF periods. We consider uncoordinated LTE-U and WiFi networks and we propose coexistence schemes that select an optimal muting period based on the saturation status of the WiFi network. In other words, the paper proposes coexistence schemes for uncoordinated LTE-U and WiFi networks which do not require a signaling protocol to exchange traffic status between the technologies. As a WiFi saturation sensing model which is capable to discriminate between saturated and unsaturated WiFi network behavior is used, the proposed coexistence scheme does not require decoding of the WiFi traffic. These features lead to considerably superior coexistence control decisions. In LTE-U and WiFi coexistence problem, the LTE-U and WiFi networks have dynamic traffic loads. For such a dynamic environment problems experience replay based reinforcement learning solutions are more efficient [9]. Hence, we propose two coexistence schemes: (a) Experience Replay (ER) based Q-learning (b) Reward selective Experience Replay (RER) based Q-learning. As uncoordinated LTE-U and WiFi networks are considered, the sensed saturation status of WiFi, which represents the traffic demand of WiFi, is stored in the experience records. In experience replay based solutions, it is recommended to use the most significant experiences for faster convergence [9]. For this reason, we also propose a RER based Q-learning coexistence scheme. In the RER based Q-learning scheme, experiences with highest reward value are used to update the Q-table. We present the performance comparison of the proposed schemes with each other and with other rule-based coexistence scheme and Q-learning based coexistence scheme which are formulated for comparison purposes. The main contributions of this work are summarized as follows:Investigation on non-coordinated coexistence of LTE and WiFi networks.Propose coexistence scheme that use ER based Q-learning and RER based Q-learning solutions for uncoordinated LTE-U and WiFi networks.Performance analysis and comparison of the proposed coexistence schemes with each other and with a rule-based and Q-learning based coexistence schemes in terms of model complexity, convergence, the accuracy of best action selection, fairness, and throughput. We are able to observe that the RER Q-learning scheme converges faster than the ER Q-learning scheme and gives 19.1% and 5.2% better aggregated throughput performance than the rule-based and Q-learning schemes, respectively. The RER Q-learning scheme also achieves 16.4% and 10.9% better fairness performance as compared to the rule-based and Q-learning schemes, respectively.

In general, this paper presents a coexistence scheme that does not require a collaboration protocol for the traffic status exchange between the co-located networks. In the proposed coexistence scheme, the LTE-U eNB uses a WiFi saturation sensing model to estimate the WiFi traffic load and selects an optimal configuration according to the WiFi saturation status. Therefore, the proposed scheme requires modifications on the LTE-U eNB side only and this makes it compatible with commercial off-the-shelf WiFi devices. This also enhances the deployment of real-time coexistence decisions as there is no delay introduced due to traffic load status exchange between the technologies. Furthermore, the proposed coexistence strategy utilizes an experience replay technique to train a model that determines the optimal configuration, making it an excellent coexistence solution for co-located networks with dynamic traffic loads.

The rest of this paper is structured as follows. Section 2 examines some recent studies on the coexistence of LTE and WiFi while Section 3 describes the definition of the problem addressed. In Section 4, the architecture of coexistence schemes in uncoordinated LTE and WiFi is described. The proposed coexistence mechanisms are discussed in Section 5. Section 6 evaluates the performance of the proposed coexistence mechanisms. Finally, Section 7 presents the conclusion of this work and outlines related future works.

Table 1 summarizes the notations used in this article.

## 2. Related Work

Many researchers have done an extensive study on the coexistence of WiFi and other networks that operate concurrently in unlicensed spectrum bands. In the next sections, we will discuss existing coexistence schemes, particularly on the solutions proposed for the fair coexistence of coordinated and uncoordinated LTE and WiFi networks.

### 2.1. Coexistence in Coordinated LTE and WiFi Networks

In [10], Almeida et al. proposed a coexistence scheme that uses blank LTE subframes to give transmission opportunities to WiFi. Simulation results in this work show that the sequence and number of the blank subframes have a significant impact on the performance of coexistence solutions. In [11], a coexistence mechanism is proposed to guarantee a fair coexistence scheme between WiFi and LTE-U. The authors proposed a mechanism that is used to adjust the LTE duty-cycle time fraction based on the traffic status of a co-located WiFi and the available licensed spectrum resource of the LTE-U.

In [12,13,14], Q-learning based CSAT mechanism that adapts LTE duty-cycle ON-OFF time ratio to the transmitted data rate is proposed. Their solution aims to maximize the WiFi and LTE-U aggregated throughput while maintaining fairness. Similarly, the authors in [15] propose a Q-learning based LTE-U and WiFi coexistence algorithm in multi-channel scenarios. By taking the idea of alternately transferring data in LTE-U and WiFi, the algorithm takes into account both the fairness and the performance of the system and optimizes the duty cycle.

In our previous work [16], Q-learning based LTE-LAA and WiFi coexistence algorithms are proposed. Q-learning is used in these algorithms to implement an autonomous selection of optimum parameter combinations that can ensure fair and efficient coexistence between co-located LTE-LAA and WiFi networks. Similarly, the authors in [17,18,19] propose a machine learning based coexistence solutions to select optimal parameters that lead to the best coexistence performance.

To ensure fair coexistence of LTE and WiFi, authors in [20] proposes a contention window (CW) size adaption algorithm-based channel access strategy. Similarly, the authors in [21] propose a mechanism for adaptively adjusting the back-off window size of WiFi and LTE duty-cycle time fraction based on the traffic status of a co-located WiFi and the available licensed spectrum resource of the LTE-U, while ensuring fair coexistence between the technologies. In [22], a reinforcement learning technique is implemented on the LTE-LAA to control its contention window size adaptively. This coexistence solution is constructed for coordinated LTE and WiFi networks. In other words, a cooperative learning method is constructed under the assumption that information across multiple systems can be exchanged.

The studies are described in this section assume that co-located LTE eNB and WiFi Access Point (AP) can exchange the exact traffic requirements between each other. In practice, the two systems do not have a dedicated common control channel that may be utilized to share traffic status. As a result, the suggested coexistence schemes necessitate a system architectural change on the two technologies to create a new channel for traffic status reporting. In other words, modifications are required in both legacy LTE eNB and WiFi AP sides and these additional system requirements make implementing coexistence schemes difficult.

### 2.2. Coexistence in Uncoordinated LTE and WiFi Networks

In this section, we discuss coexistence mechanisms that consider uncoordinated LTE and WiFi networks. In uncoordinated LTE and WiFi networks, there is no cooperative channel used to exchange the traffic status between the technologies. Hence, additional features that enable to sense the WiFi traffic have to be included in the co-located technologies.

In [23,24], number of WiFi APs are determined to estimate the WiFi traffic. The work in [23] proposes a method for discriminating between one and two WiFi APs by employing an auto-correlation function on the WiFi preamble and setting appropriate detection thresholds to estimate the number of active WiFi APs. Similarly, an energy detection-based approach is used to distinguish between one and two WiFi APs in [24]. The work in [25] offers an ML-based strategy for determining numerous WiFi APs using energy levels measured. This work also proposes an LTE-U/WiFi coexistence scheme that utilizes the number of APs detected by the system. Generally, the proposed approach of determining the number of co-located WiFi APs is a less complex approach to get a rough estimate of the WiFi traffic as compared to installing a full WiFi receiver at the LTE eNB side to decode the WiFi packets. The problem in using the number of active WiFi APs to estimate Wifi traffic is the fact that each active WiFi AP can have a varied traffic load. Hence, counting the number of active APs can lead to wrong coexistence decisions.

In our previous work [26], the CNN model that can identify the duration of each transmitted frame from each co-located technology was proposed and validated. The validation was carried out utilizing commercially available LTE and WiFi hardware. Similarly, CNN based models are used to perform identification of WiFi transmissions from other co-located transmissions of other technologies in [27,28]. Our CNN based technology classification proposed in [26] is used to implement coexistence schemes between private LTE and WiFi in [29,30]. These coexistence schemes use Channel Occupancy Time (COT) to estimate the WiFi traffic and select optimal transmission time of LTE and WiFi. However, COT is not a good indicator of the WiFi traffic load as it depends on the packet size and the number of active users [8]. Hence, COT based coexistence decisions are not efficient in variable packet size and active nodes.

The authors in [31] propose a coexistence solution that uses reinforcement learning to estimate WiFi traffic demand. In this work, the LTE eNB uses reinforcement learning to learn and predict future WiFi traffic demands. Similarly, the authors in [32] offer a Q-learning-based methodology to estimate WiFi traffic characteristics. The suggested solution provides a decision-making framework that uses carrier detection at the LTE eNB to determine WiFi idle time. The primary goal of this effort is to maximize unlicensed LTE exploitation of idle spectrum resources. In general, these systems estimate WiFi traffic demand by continuously monitoring the average number of total idle slots, the average number of successfully sent WiFi packets, and the average number of collisions, all of which can indirectly indicate WiFi traffic demands. However, the proposed solutions only consider unsaturated WiFi networks. In practice, the co-located WiFi network can be either saturated or unsaturated and the metrics found in the DRL-based WiFi traffic estimation, such as the number of successfully transmitted WiFi packets, differ in saturated and unsaturated WiFi traffic [33].

In [22], reinforcement learning techniques are used to tune the contention window size for both LTE-LAA and WiFi nodes. This coexistence solution is constructed for uncoordinated LTE-LAA and WiFi networks. Reinforcement learning based solutions are implemented in both LTE and WiFi nodes. This non-cooperative version is designed for better practicability, and it is demonstrated that the proposed learning method can significantly improve total throughput performance while maintaining fairness. Even though there is no need for a cooperative channel that is used for information exchange between the technologies, this solution requires modifications in both LTE eNB and WiFi AP nodes.

### 2.3. Enhancements

We reviewed different coexistence schemes that are proposed to achieve fair and efficient coexistence of LTE and WiFi networks. However, most of the proposed solutions consider coordinated LTE and WiFi networks in which the traffic status of the technologies is exchangeable via a collaboration protocol. There are also some papers that propose WiFi traffic demand estimation techniques used in uncoordinated coexistence schemes. However, WiFi traffic load estimation is mostly done based on wrong assumptions such as the number of active APs, or COT. Therefore, this paper proposes an efficient coexistence scheme for uncoordinated LTE and WiFi which requires modifications only on the LTE eNB side. To estimate the traffic demand of WiFi, the coexistence scheme uses the saturation sensing model proposed in our previous work [8]. Moreover, we have reviewed many coexistence schemes that use Q-learning to determine optimal configuration. However, Q-learning is not an efficient solution for problems with dynamic environments. For this reason, we propose coexistence schemes that utilizes the experience replay technique to cope with the dynamic nature of traffic loads of the co-located technologies in coexistence problems. In general, this paper proposes a coexistence scheme that (i) only requires modifications on the LTE eNBs, i.e., no modifications are required on the legacy WiFi nodes (ii) does not require to decode WiFi traffic, (iii) does not require any coordination signaling protocol between the technologies, and (iv) is a suitable coexistence solution for co-located networks with dynamic traffic loads as it uses an experience replay technique for the optimal configuration selection.

## 3. Problem Definition

In this work, we aim to propose a fair and efficient coexistence scheme for uncoordinated LTE-U and WiFi networks. In our previous work [8], we have developed a CNN-based solution that classifies saturated and unsaturated WiFi networks. The developed solution can be used by advanced coexistence schemes that aim to achieve fair and spectrum efficient coexistence between technologies co-located with WiFi. In this work, we aim to develop a coexistence scheme that is used for autonomous selection of optimal LTE-U ON and OFF duration ratio which maximizes aggregated throughput of the two technologies and fairness between the technologies.

The main goal of the algorithm is to select an optimal LTE-U ON and OFF duration ratio based on LTE-U and WiFi network traffic loads. The LTE-U estimates the traffic demand of the WiFi network based on the saturation sensing model and classifies the WiFi networks as either saturated or unsaturated based on their offered traffic load. A WiFi network is considered as unsaturated if its aggregated throughput has not reached the maximum system throughput limit and saturated otherwise [33]. Similarly, for the LTE network, LTE offered traffic load ($LTE_{\mathrm{Of}}$), the target throughput ($LTE_{\mathrm{Tar}}$), and obtained throughput ($LTE_{\mathrm{Ob}}$) are considered to select the optimal LTE-U ON and OFF duration ratio. $LTE_{\mathrm{Ob}}$ refers to the throughput obtained by the LTE network when a certain TXOP is selected and $LTE_{\mathrm{Tar}}$ is set as the offered load throughput if the offered data rate is less than the saturation throughput of the LTE network. On the other hand, $LTE_{\mathrm{Tar}}$ is set as the LTE saturation throughput if the offered data rate is greater than the saturation throughput of the LTE network. Note that the LTE saturation throughput is the maximum capacity of the LTE network in a standalone LTE network. In other words, the saturation throughput is the maximum aggregated capacity of all the eNBs of the LTE network for all the User Equipments (UEs) when the LTE uses the whole spectrum.

The goal of the coexistence scheme is to select an optimal LTE-U ON and OFF duration ratio so that the WiFi network remains unsaturated while the $LTE_{\mathrm{Ob}}$ is close to $LTE_{\mathrm{Tar}}$, if possible. Otherwise, the LTE-U ON and OFF duration ratio has to be selected in such a way that the aggregated throughput and fairness of the two technologies are maximized. In this article, we aim to achieve these objectives considering the following constraints: (a) no collaboration protocol is used to exchange the traffic status between the technologies (b) the coexistence is implemented on the LTE-U eNB side and there is no modification required in the WiFi network’s commercial devices.

## 4. System Model

In this section, we propose and discuss the architecture of the coexistence scheme for uncoordinated LTE and WiFi networks that use WiFi saturation status to estimate the traffic demand of the WiFi network. Figure 1 shows the main blocks of the coexistence schemes implemented in the LTE-U eNB considering uncoordinated LTE-U and WiFi networks. The LTE eNB has three main features, namely the *Technology recognition model*, the *WiFi saturation sensing model*, and the *Coexistence decision model*. The *Technology recognition model* is used to capture and identify concurrent transmissions by different co-located technologies. In practical applications, the LTE and WiFi transmissions are classified based on technology recognition solution that requires capturing and processing the IQ samples of LTE and WiFi traffic [26]. Once the WiFi frames are classified, they are fed into the *WiFi saturation sensing model*, where the saturation state of WiFi is determined based on the CNN model proposed in our previous work [8]. The *WiFi saturation sensing model* is used to distinguish saturated and unsaturated WiFi traffic in real-time by analyzing the Inter-Frame Spacing (IFS) distribution. The IFS distribution is derived by processing the statistics obtained from the *Technology recognition model*. Finally, the *Coexistence decision model* is used to decide the optimal action that leads to efficient coexistence. This decision is made based on the saturation status of the WiFi network and the status of the LTE-U network. The proposed coexistence decision algorithms are discussed in the next sections.

Figure 1 shows that the coexistence scheme for uncoordinated LTE-U and WiFi networks is executed by the central eNB. The LTE central eNB uses technology recognition system [26] and a WiFi saturation sensing model [8] to estimate the traffic demand of the co-located WiFi network. The decision made by the central eNB also considers the number of co-located eNBs, $LTE_{\mathrm{Of}}$, $LTE_{\mathrm{Tar}}$, and $LTE_{\mathrm{Ob}}$ of the collocated eNBs within its collision domain. The central eNB collects this information using the X2 interface and the X2AP services [34]. Once the best configuration is determined by the central eNB, this X2 interface is used to send the decision to the co-located eNBs. After that, each eNB adjusts its transmission time based on the decision. A similar approach of coexistence decision exchange between central eNB and other co-located eNBs is also used in existing LTE/WiFi coexistence solutions [13].

### Assumptions

In the proposed coexistence solutions, we consider coordinated eNBs which exchange coexistence decision made by a central coordinator eNB. In terms of simulation, the parameter parsing is done by functions and receiving the updated configurations from the central node takes negligible latency and overhead. However, for real-world deployments, this communication between nodes can be accomplished by combining the X2 interface and the X2AP services [13]. The X2 is the designation of the interface that connects one eNB to another as defined by 3GPP in [34]. Similarly, eNBs in different operators can exchange the information through corresponding Mobility Management Entity (MME) via S10 interface [35].

In practical applications, the LTE and WiFi frames are classified in *Technology recognition model* based on technology recognition solution as proposed in our work in [26]. In the context of this article, we use the ns-3 simulator to model the LTE and WiFi networks and the technology recognition step is emulated by generating log files that represent statistics of LTE and WiFi frames.

To minimize the computational complexity, a coexistence scheme has to report the optimal ON-OFF duty-cycle ratio every prediction duration T seconds, where T is an optimally selected network traffic prediction period based on the traffic dynamics of the co-located networks. In practice, the traffic dynamics of the co-located networks have to be analyzed and the value of prediction duration T has to be set low enough to match the traffic dynamics of the WiFi network [36]. However, the selection of optimal time T is beyond the scope of this study.

## 5. Proposed Coexistence Solutions for Uncoordinated LTE-U and Wi-Fi Networks

The ultimate goal of this article is to propose a WiFi saturation based fair and efficient coexistence scheme for uncoordinated LTE and WiFi networks. In this section, we formulate and describe coexistence schemes for non-coordinated coexistence solutions. For comparison and validation purposes, we formulated simple rule based and classical Q-learning coexistence schemes. These coexistence schemes are formulated in such a way that they can be utilized for uncoordinated LTE-U and Wi-Fi networks. For this reason, the first two subsections of this section describe a simple rule based adaptive coexistence scheme and Q-learning based coexistence scheme for uncoordinated LTE-U and WiFi networks. The proposed ER and RER based Q-learning coexistence schemes are described in the last two subsections of this section.

### 5.1. Rule Based Coexistence Scheme

In this subsection, we describe a rule-based coexistence scheme that is used to determine the number of LTE blank subframes ($N_{b}$) that lead to the highest fairness and spectrum efficiency. This rule based coexistence scheme is formulated for comparison purposes. The main goal of the algorithm is to select a $N_{b}$ value based on *LTE*_State_ and *WiFi*_State_. The *WiFi*_State_ represents WiFi state which can be either saturated (*W*_S_) or unsaturated (*W*_U_). $LTE_{\mathrm{State}}$ represents the state of LTE which is determined based on $LTE_{\mathrm{Tar}}$ and $LTE_{\mathrm{Ob}}$. The algorithm uses possible states of LTE shown in Table 2. When blank subframes are introduced in the LTE frame, the LTE throughput decreases proportionally [30]. Hence, the 10 states in Table 2 are used to represent all the possible range of throughput levels.

Algorithm 1 shows the steps of the proposed rule-based coexistence scheme. Initially, 50% of the LTE subframes are set as blank ensuring equal spectrum share for both technologies. Then, the fairness of this action is evaluated. We consider the action as 100% fair if the LTE network achieves sufficient throughput (L_10_) and the WiFi network is unsaturated. We also consider the action as 100% fair if $LTE_{\mathrm{State}}\neq L_{10}$ and the WiFi network is saturated when the technologies are set to share the spectrum equally with 50% blank subframes. On the other hand, if the action is not fair, the value of *N*_b_ is adjusted based on LTE and WiFi states. If the WiFi network is saturated and the LTE network achieves sufficient throughput (*L*_10_), the number of blank LTE subframes is increased so that WiFi gets more spectrum thereby leading to better fairness. Contrarily, if the WiFi network is unsaturated and the LTE network does not achieve sufficient throughput, i.e., $LTE_{\mathrm{State}}\neq L_{10}$ the number of blank subframes are reduced to achieve better fairness. $N_{t}$ represents the number of actions executed until the best action. This process has to be repeated periodically to adapt to the traffic dynamics of the co-located networks.

### 5.2. Q-Learning Based Coexistence Scheme

Reinforcement Learning (RL) is an area of machine learning where an agent learns from its interaction with the environment. Q-learning is a type of reinforcement learning algorithm that implements dynamic programming and Monte Carlo concepts to solve sequential decision-making problems in the form of Markov Decision Processes (MDP) [37]. In MDP, the agent interacts with the environment and makes decisions based on the environment states. The agent chooses actions and the environment on the other hand is used to update the responses to these actions in the form of observations or new situations. The agent aims to maximize the cumulative reward of taking an action in a particular state by learning a policy π from trial and error in the environment. In the process of learning iteration, the agent initially observes the current state *s* on the environment. The agent then selects the corresponding action *a* according to the policy π and it receives the reward value *R* from the environment. After the action is taken on the environment the state changes to $s^{\prime }$. Finally, the optimal policy π* is updated based on the current state and the reward value [37].
**Algorithm 1:** Rule based coexistence scheme
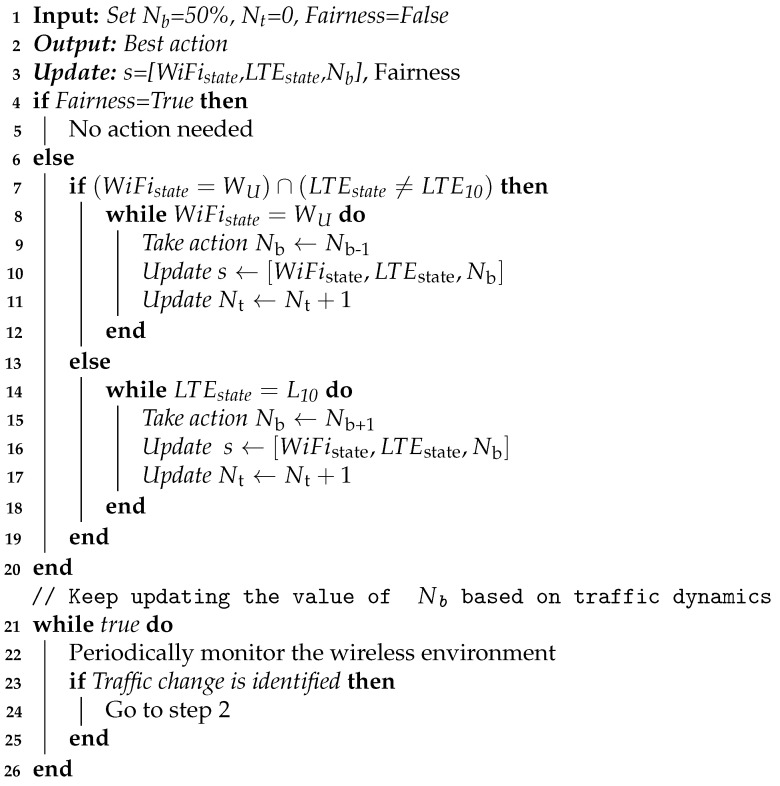


In the Q-learning process, the agent balances between exploration and exploitation while it tries to find an optimal strategy in the selection of action *a* after observing the current state *s* of the environment. In the exploration case, the agent selects an action randomly, expecting that the random action will lead to a higher cumulative reward in the next iterations. On the contrary, in the exploitation process, an action is selected based on the latest expected optimal policy. Generally, during exploitation, the agent uses the experience from already investigated actions and selects the optimal action based on them, while during exploration, the agent investigates and experiences new actions. ε-greedy policy is a policy used to balance exploration and exploitation by using a decision value ε. The decision value is selected in the range 0≤ε≤1 and it is used to decide whether the agent will explore or exploit in every step. The agent uses exploration and exploitation with a probability of ε and 1−ε, respectively.

Algorithm 2 shows how Q-learning can be used for optimally selecting the number of blank subframes (*N*_b_), aiming at maximizing fairness and the aggregated throughput of LTE and WiFi networks. The procedures show how the agent selects an optimal value of *N*_b_ based on LTE throughput status and WiFi saturation status. The algorithm also shows how exploration and exploitation are balanced by an adjustable value of ε. Initially, the ε is set to a value close to 1 so that the agent starts with a higher probability of exploration so that it quickly explores different states. After $N_{\varepsilon }$ number of iterations, the value of ε is reduced by a factor of $f_{\varepsilon }$, until a minimum value of $\varepsilon_{\min}$ is reached.

The key elements of the Q-learning process for optimal *N*_b_ selection can be described as follows:**Agent:** eNB, we assume the LTE-U eNBs belong to the same operator which cooperates together.**State:** The state at time t+1 is determined by the status of the environment after an action *a* is taken at time *t*. The statuses of the WiFi and the LTE networks are used to represent the state after a frame configuration of LTE is selected.The new state $s^{\prime }$ can be represented as:
(1)$$s^{\prime }=\left[WiFi_{\mathrm{State}},LTE_{\mathrm{State}},N_{b}\right].$$*WiFi*_State_ can be saturated (*W*_S_) or unsaturated (*W*_U_). *LTE*_State_ represents the state of LTE which is determined based on LTE target throughput and LTE obtained throughput. The state of the WiFi network is represented by its saturation status which is determined based on our previous work [8]. On the other hand, the state of the LTE network is determined by the *LTE*_obtained_ and *LTE*_Target_ throughput. The possible states of *LTE* are shown in Table 2. *N*_b_ is the number of blank LTE subframes in a single frame. The state-space *S* includes all combination of *WiFi*_State_, *LTE*_State_, and *N*_b_ values.**Action:** The action *a* at time *t* represents the selection of $N_{b}$ number of blank LTE subframes.**Reward:** LTE network is required to attain the highest possible throughput by accessing the spectrum while maintaining fairness with WiFi. However, the performance of the WiFi can not be directly accessed by the agent as our goal is to propose a coexistence scheme that does not require signaling exchanges between the WiFi AP and LTE eNB. For this reason, we consider the CNN-based WiFi network saturation sensing model to determine the status of the co-located WiFi. The reward value for action *a* taken at a state *s* is computed based on the newly achieved state $s^{\prime }$ as follows:
(2)$$r_{Q}=\left\{\begin{array}{cc} 100 & \;\mathrm{if}\;s^{\prime }\;=\;\left([W_{U},L_{10},N_{b})\right]\mathrm{or}\left(\left[W_{S},LTE_{\mathrm{State}}\neq L_{10},5\right]\right)\\ 100*\rho *\beta *\frac{N_{b}}{10} & \mathrm{otherwise} \end{array}\right..$$
where *N*_b_ is the number of blank LTE subframes and β is used based on the values in Table 2. ρ is 1 if WiFi state is *W*_U_ and 0.5 if WiFi state is *W*_S_. The value of β increases as the *LTE*_Obtained_ gets close to *LTE*_Target_. Similarly, the value of ρ is higher when the WiFi network is unsaturated. Equation (Equation 2) shows that the maximum reward is obtained when LTE gets the highest possible throughput range (state *L*_10_) and WiFi is unsaturated (*W*_U_). Similarly, the highest reward can be achieved if both technologies are spectrum hungry while both technologies are given equal spectrum share, i.e., if LTE is not at state *L*_10_ and WiFi is at state *W*_S_ while *N*_b_ = 5. Otherwise, the reward function depends on the values of β, ρ, and *N*_b_. As *N*_b_ increases, LTE gets a lower spectrum share and hence the value of β decreases. On the other hand, as *N*_b_ increases, there is a higher probability that the WiFi gets sufficient spectrum to attain unsaturated state leading to ρ = 1. Generally, the values of β and ρ are selected such that an action with a higher aggregated throughput and fairness leads to a higher reward value.

### 5.3. Experience Replay Based Q-Learning

The traffic load of wireless networks is mostly dynamic. In such a dynamic environment state–action pairs are not consistent over time. This makes predicting future rewards more complex. For this reason, the Q-learning algorithm is mostly applied in stationary environments where state-action pairs are consistent. However, this problem can be solved by using Experience Replay (ER). ER is an approach used to minimize action–state pair oscillation. This is achieved by storing a large number of past experiences. A single stored experience is represented by a combination of the current state, action, reward, and next state ($s,a,r,s^{\prime }$). In traditional Q-learning, the Q-table is updated based on the single latest experience. However, Q-table values are updated by taking a random portion of the buffered experiences in case of experience replay-based Q-learning [9].
**Algorithm 2:** Q-learning based algorithm for dynamic selection of *N*_b_
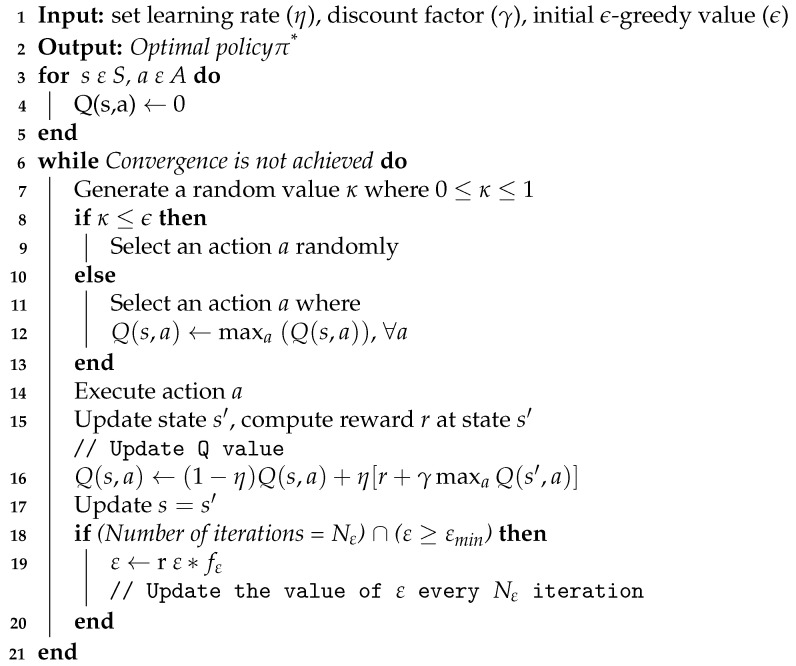


In ER-based Q-learning, $N_{E}$ number of experiences are recorded as a single batch from the experience buffer. The batch size remains fixed as newly recorded experiences keep pushing the oldest experience out of the list. The use of such an experience record enables the agent to learn more from important individual experiences, i.e., some important experiences can be used multiple times to achieve faster convergence. The importance of the reward can be defined based on different criterion such as it corresponding reward or time of occurrence. The experience record also helps to recall rare occurrences and harmful correlations. For this reason, ER is a promising approach to achieve an optimal policy from fewer experiences. Hence, we propose an ER-based Q-learning solution for LTE and WiFi network coexistence.

The agent, action, and state elements of ER-based Q-learning are similar to the traditional Q-learning based solution described in Section 5.2. In traditional Q-learning, we can only estimate the reward value of a given action based on the current state *s* and the new state $s^{\prime }$ after the action is taken. However, this equation only shows how good the action is in this state but with the knowledge of a single experience, we can not determine if the action is the best action among all the possible actions. For this reason, we propose ER based Q-learning solution to cope with the dynamic nature of the traffic load of the LTE and WiFi networks. For the dynamic traffic environment, we need to record the experiences by taking all possible actions for each traffic load combination and these observed experiences can be used to determine which action is the best. For each experience, a fairness factor ($f_{f}$) is included to indicate the fairness level in the experience. In other words, $f_{f}$ value is selected in such a way that its value increases with the fairness level of the action. $f_{f}$ is generated using the following equation:(3)$$f_{f}\left(a\right)=\left\{\begin{array}{cc} \rho *\beta & \mathrm{if}\;\mathrm{WiFi}\;\mathrm{state}\;\mathrm{is}\;W_{U}\\ \rho *\beta *\frac{N_{b}}{10} & \mathrm{otherwise} \end{array}\right..$$
where *N*_b_ is the number of blank LTE subframes and β is used based on the values in Table 2. ρ is 1 if WiFi state is *W*_U_ and 0.5 if WiFi state is *W*_S_. The fairness factor equation shown in Equation (Equation 3) is set in such a way that its value increases as the two technologies utilize a fair share of the spectrum. As far as the WiFi the network is unsaturated, *W*_U_ the fairness factor value increases when the LTE network’s obtained throughput increases. In the equation, the LTE network obtained throughput is reflected by the value of β. On the other hand, when the WiFi network gets saturated, the value of the fairness factor depends on the LTE obtained throughput (reflected by β) and *N*_b_. Generally, the equation leads to highest fairness factor value if *L*_10_ and WiFi is at state *W*_U_ are obtained at a given action. However, if both LTE and WiFi networks have a higher traffic load, these *L*_10_ and *W*_U_ state can not be achieved simultaneously in all possible actions. For this case, the highest fairness factor is obtained when both technologies share the spectrum equally, i.e., *N*_b_ = 5.

In case of experience replay based Q-learning, all possible actions are observed for each given traffic load. All actions are recorded with their corresponding *WiFi*_state_ and *LTE*_state_. The value of $f_{f}\left(a\right)$ is then computed based on the LTE and WiFi states and the corresponding reward values are computed using:(4)$$r_{\mathrm{ER}}=\left\{\begin{array}{cc} 100 & \mathrm{if}\;F(a)\;\mathrm{is}\;\max_{a}F(a),\;\forall a\\ 0 & \mathrm{otherwise} \end{array}\right..$$

The reward function presented in Equation (Equation 4) leads to the highest value when an action taken has the maximum possible fairness factor $f_{f}\left(a\right)$ value as compared to all the other possible actions recorded as experiences. Without the use of an experience record, it is not possible to determine if a given action is the best action as compared to other possible actions. For this reason, the value of $f_{f}\left(a\right)$ can not be used in the classical Q-learning reward function. In a dynamic environment, the best action can only be certainly determined after observing and recording all the possible actions. This is the reason behind proposing an ER based Q-learning solution for the LTE-U and WiFi network coexistence problem in a dynamic environment.

Algorithm 3 shows how the ER based Q-learning can be used to predict an optimal policy π*. The procedures show how $N_{E}$ recorded experiences are used to update the Q-network. The dynamic behavior of the environment is represented by generating random traffic load on the LTE and WiFi network. For a given fixed traffic load all possible actions are taken and values $\left\{s,a,f_{f},s^{\prime }\right\}$ are recorded in $R_{f}$ for every action. Once the values of $R_{f}$ are stored for all actions, the reward values are computed and the values $\left\{s,a,r,s^{\prime }\right\}$ stored to $R_{f}$ for every action *a*. These values are then stored to $R_{E}$ until the number of experiences reaches $N_{E}$. Once $N_{E}$ experiences are recorded, each stored element $\left\{s,a,r,s^{\prime }\right\}$ is used to update the Q-network based on the procedures mentioned in Algorithm 2. This whole process is repeated until the Q-matrix converges.

### 5.4. Reward Selective Experience Replay Based Q-Learning

We have seen that the ER-based Q-learning solution stores experience $\left\{s,a,r,s^{\prime }\right\}$ for all actions until the number of experiences reaches $N_{E}$. After recording $N_{E}$ experiences, each stored element $\left\{s,a,r,s^{\prime }\right\}$ is used to update the Q-network using the steps described in Algorithm 3. In most circumstances, it is not recommended to use every experience in the database, so we must specify some sort of selection process. For this reason, we also investigate RER based Q-learning strategy in which we keep only the $N_{E}$ best experiences with the highest attained reward in each action. In other words, the reward for each action is calculated using Equation (Equation 4), and the corresponding experience is used to update the Q-matrix values if the acquired reward is 100.

In the RER based Q-learning, the main procedures used to update the Q-table are similar to the procedures used in ER based Q-learning scheme depicted in Algorithm 3. The difference between ER based Q-learning scheme and RER based Q-learning scheme is, in ER based Q-learning scheme every recorded experience is used to update the Q-table while RER based Q-learning scheme updates the Q-table based on experiences with best reward value. This reward based experience record update is adopted from [9].
**Algorithm 3:** ER Q-learning based algorithm for dynamic selection of *N*_b_
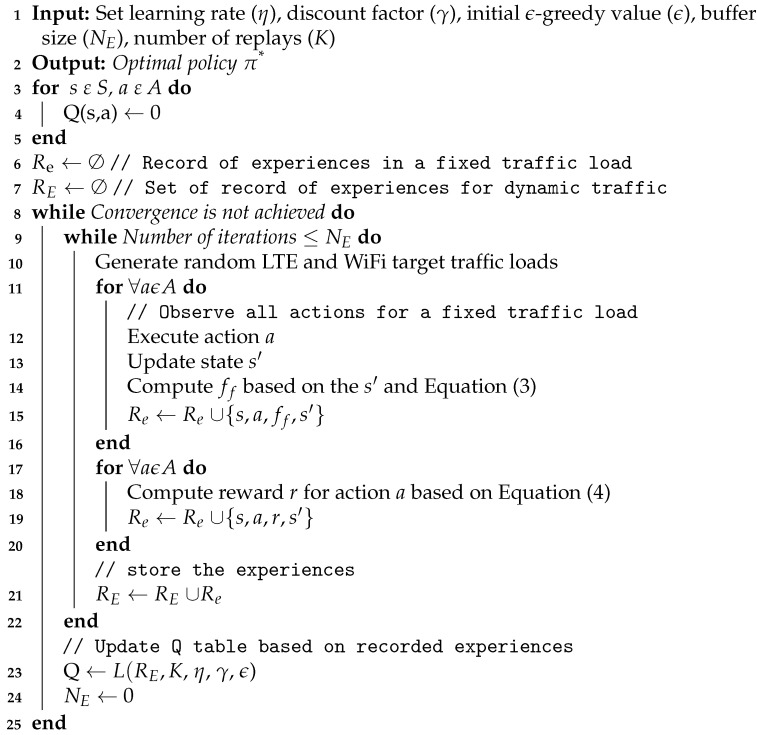


## 6. Performance Evaluation

### 6.1. Evaluation Scenario

Before going to the performance analysis of the proposed coexistence schemes we start by defining and describing the evaluation scenario. In [38], 3GPP proposed evaluation scenarios that can be used to evaluate the coexistence of WiFi and LTE-U networks. In our case, we use the indoor (Hotspot) scenario which is one of the proposed evaluation scenarios.

Figure 2 presents the 3rd generation partnership project (3GPP) indoor scenario considered in this work which is composed of two operators, called operator A (LTE-U) and operator B (WiFi). Each operator deploys four small cells operating in a rectangular room with dimensions of 50 × 120 m. The room has no walls and the four base stations of each operator are equally spaced in the X-axis. Each cell is connected to *n* UE/stations, where *n* is randomly selected between 1 and 5. The UE/stations are randomly located within the room. The distance between base stations from different operators is 5 m while the distance between base stations of the same operator is 25 m.

We implement and analyze the proposed coexistence schemes using the ns-3 simulator as it is one of the prominent simulators to model wireless networks. The simulator is also convenient to model coexistence scheme as it can be used to generate dynamic traffic loads of LTE-U and WiFi networks. Hence, we use ns-3 version specifically released with LTE/WiFi coexistence implementations [39]. This ns-3 simulator release is used to model the LTE-U and WiFi networks and the proposed coexistence scheme.

The simulation parameters used to model the co-located technologies are shown in Table 3. During the evaluation and testing of the proposed solutions, each LTE UE or WiFi station generates a random offered User Datagram Protocol (UDP) data rate. The UDP data rates are randomly picked from {50 kbps, 500 kbps, 1 Mbps, 2 Mbps, 4 Mbps}. Once a certain UDP data rate is selected by a UE, the specific load will be held constant for some specific duration $T_{D}$ before it is updated by another new offered data rate. The duration time where the offered data rate remains constant is randomly picked from an interval between 5–15 s. In this way, there will be from 20 to 60 changes in data rate (total simulation duration divided by minimum and maximum possible values of $T_{D}$) in a total of 300 s simulation time. These values used for simulation of the dynamic traffic are adopted from [13].

The considered network uses a constant speed propagation delay model named as *ConstantSpeedPropagationDelayModel* and a propagation loss model named *ITU InH* [40,41]. In the coexistence schemes, the IEEE802.11n mode is used to model the WiFi network. Packet size of 1500 bytes, minimum contention window of 15, and maximum contention window of 1023 are used for the WiFi network. Furthermore, slot time, Short Inter-Frame Spacing (SIFS), and DCF Inter-Frame Spacing (DIFS) are 9 μs, 16 μs, and 34 μs, respectively.

A simulation scenario modeled in ns-3 can be integrated with the RL models using the framework described in [42]. In our simulations, this approach is used to integrate the proposed RL models with the ns-3 simulator. The ns-3 simulator is used as an environment that generates dynamic simulation scenarios reflecting the dynamic traffic load variations in the LTE-U and WiFi network models. The RL agent selects an optimal action using the rule-based, Q-learning, ER Q-learning, and RER Q-learning schemes, which are implemented using Python 3.9.0. In the coexistence schemes, the WiFi saturation status sensing model is used to estimate WiFi network saturation status. This WiFi saturation sensing model is a neural network algorithm framework developed in our previous work [8], which is implemented on Python 3.9.0 with Tensorflow 1.1.0 and Keras 2.2.5.

### 6.2. Convergence and Complexity Analysis

In this section, we will discuss the convergence and complexity analysis of the coexistence schemes described in Section 5. We have described that all the schemes use the saturation state of WiFi, which is determined based on our previous work [8]. The WiFi saturation status is determined based on inter-frame statistics collected from wireless signals measured in 1 s. In this previous work, we have seen that the saturation status classification of WiFi traffic in a duration of 1s practically requires an end-to-end processing time of 1.032 s. This time is the sum of time required for capturing I/Q samples, pre-processing and classifying process (which requires 951 ms [26]), and time required to classify the saturation status of the WiFi network (which requires 81 ms [8]).

The first coexistence scheme described in Section 5 is the rule-based solution. The rule-based solution is used to select an optimal value of $N_{b}$ that leads to fairness based on Algorithm 1 for 20,000 randomly generated traffic load values of LTE-U and WiFi networks. The dynamic traffic loads are generated by running the 300 s simulation multiple times and varying the traffic based on the approach described in Section 6.1. In this scenario, the average value of $N_{t}$ was found to be 2.68, i.e., an average of 2.68 actions are executed until fairness is achieved for each fixed traffic load.

Using the rule-based approach, the total time required to report an LTE frame configuration in each trial is 1 s (wireless signal measurement) + 1.032 s (processing time of technology recognition classification + saturation sensing) + (rule-based decision − negligible processing time) = 2.032 s. As the best action requires 2.87 trials on average, the best action requires a period of 2.87 × 2.042 s = 5.832 s. Basic challenges in using the rule-based approach include:It is not convenient for co-located networks that have very dynamic traffic as determining the best action requires an average time of 5.832 s.Each trial is a new configuration and it affects the performance of the technologies as far as it is not the best one. i.e., even though the traffic remains constant for a long time, the best action is found after 5.832 s and the performance remains poor as wrong configurations are used in the meantime until the best action is not finally taken.Rule-based solutions are not easily scalable if the number of possible actions is increased and this will further increase the time required to find the best action.

The second coexistence scheme adopted for uncoordinated LTE-U and WiFi networks is the Q-learning based solution described in Section 5.2. The Q-learning parameters η and γ were set to 0.75 and 0.85, respectively. Similarly, the initial value of ε is set to 0.9, while $N_{\varepsilon }$ is set to 500. After 500 iterations, the value of ε is reduced by a factor of $f_{\varepsilon }$ = 90%, until a minimum value of $\varepsilon_{\min}$ = 0.05 is reached. These values were selected by an inspection as they lead to the best convergence. The convergence of the Q-learning based solution is shown in Figure 3. We can see that a large number of iterations (about 184,000) are necessary before the Q-matrix converges. This is because wireless network traffic is mostly dynamic, and state–action pairs are not consistent over time.

Figure 3 also shows that using ER-based Q-learning can lead to faster convergence. The number of experiences per batch ($N_{E}$) was set to 500, while the number of experience replays (*K*) was set to 4. These values were also chosen by inspection because they lead to the best convergence, while the remaining hyper-parameters were left unchanged from the Q-learning based scheme. Figure 3 also shows that using ER of experiences with the best reward leads to even faster convergence. This is also expected as experiences with the best outcomes are selected and replayed instead of investigating every experience.

One prevalent issue with machine learning-based solutions is that they require a learning phase before providing an optimal solution. The learning phase is a computational cost of the Q-learning process, as is the case with other learning methods. During this phase, an agent investigates several possible actions in each potential condition to learn about the environment. In the case of using experience relay, the training part requires registering and reading experiences. However, after the environment has been learned, the best action in every given condition can be taken, resulting in the optimal solution. For a given state *s*, Q-learning based selection of optimal action *a* requires an average time of 0.018 ms (average over 100 runs). In general, the total time required to report an optimal action requires 1 s (wireless signal measurement) + 1.032 s (processing time of technology recognition classification + saturation sensing) + 0.018 ms (Q-matrix based decision ) ≈ 2.032 s. This time is the processing time required in real-time applications, as it includes the time required by the technology recognition solution [26] to capture, pre-process, and classify WiFi frames on the medium. This means our proposed scheme can estimate the traffic and report an optimal action every 2.032 s. This interval is practically suitable as accurate traffic load prediction of real-time applications can be achieved even with a higher prediction interval (up to 5 s) [36].

However, we employ a discrete event simulator (ns-3) in our simulation, and parameter parsing is done via functions, thus the collection of WiFi frame statistics occurs with negligible processing time. On the other hand, the proposed coexistence schemes do not necessitate information exchange between the LTE and WiFi networks and this makes the solutions less complex as there is no need to make any modifications to the WiFi network elements.

### 6.3. Fairness and Throughput Performance

In this section, we present the performance of the schemes presented in Section 5 in terms of the fairness and aggregated throughput obtained in our simulations. The aggregated throughput is used as a performance evaluation metric as it reflects the spectrum efficiency of the coexistence schemes. The corresponding LTE-U and WiFi throughput values, on the other hand, are used to assess the fairness of the coexistence schemes. During the performance evaluation of the coexistence solutions, each LTE UE or WiFi station generates random data rates, which are randomly picked between {50 kbps, 500 kbps, 1 Mbps, 2 Mbps, 4 Mbps}. The UDP data rate of UE/station is updated after a certain duration $T_{D}$. The value of $T_{D}$ is randomly picked from 5–15 s, leading to 20 to 60 possible fixed load cases in a total duration of 300 s simulation time. To achieve statistical regularity, this 300 s of simulation time is repeated for 100 iterations and, fairness and aggregated throughput are computed for each fixed load.

The aggregated throughput in each fixed load duration is the sum of WiFi obtained throughput and LTE obtained throughput whereas the corresponding fairness value is given by:(5)$$f_{\mathrm{ev}}=1-\left|\frac{WiFi_{\mathrm{Tar}}-WiFi_{\mathrm{Ob}}}{LTE_{\mathrm{Ref}}}-\frac{LTE_{\mathrm{Tar}}-LTE_{\mathrm{Ob}}}{LTE_{\mathrm{Ref}}}\right|.$$
where $LTE_{\mathrm{Ref}}$ and $WiFi_{\mathrm{Ref}}$ are the reference throughput values which are the maximum possible aggregated throughput values that can be attained when the LTE and WiFi networks operate in standalone mode. $WiFi_{\mathrm{Tar}}$ is the target throughput of the WiFi network and it is equal to the offered data rate of the WiFi network as far as the offered data rate of the WiFi network does not exceed the $WiFi_{\mathrm{Ref}}$. On the other hand, the target throughput is set equal to $WiFi_{\mathrm{Ref}}$ if the offered data rate of the WiFi network is greater than the $WiFi_{\mathrm{Ref}}$. The target throughput of the LTE network ($LTE_{\mathrm{Tar}}$) is also set in a similar way based on the offered data rate and reference throughput ($LTE_{\mathrm{Ref}}$) of the LTE network. $WiFi_{\mathrm{Ob}}$ and $LTE_{\mathrm{Ob}}$ are the obtained throughput values of the WiFi and LTE networks, respectively.

Equation (Equation 5) is used to evaluate the fairness of an action taken at a given state. This fairness value considers the obtained and target throughput values for the LTE and WiFi networks [14,38]. The fairness evaluation equation is set in such a way that the fairness of a given action is high if the values of $\frac{{WiFi}_{\mathrm{Tar}}-{WiFi}_{\mathrm{Ob}}}{LTE_{\mathrm{Ref}}}$ and $\frac{LTE_{\mathrm{Tar}}-LTE_{\mathrm{Ob}}}{LTE_{\mathrm{Ref}}}$ are close. Otherwise, if these two values have a higher gap, it indicates that one network is getting higher spectrum share and hence the fairness of a given action decreases. In other words, the fairness of an action increases if both technologies are able to achieve close fraction *X*, where *X = (target throughput − obtained throughput)/(reference throughput)* for each technology.

Figure 4a shows the aggregated throughput of rule-based, traditional Q-learning based, ER Q-learning based, and RER Q-learning based solutions. The bar graph shows that the ER Q-learning-based and RER Q-learning-based coexistence schemes lead to a better coexistence in terms of the obtained aggregated throughput. On the other hand, the rule-based and traditional Q-learning-based solutions have relatively lower performance.

Figure 4b shows the Cumulative Distribution Function (CDF) of aggregated throughput during the simulation. The CDF shows the distribution of aggregated throughput values obtained in each fixed traffic load duration $T_{D}$ during the simulation period. For example, the probability of getting an aggregated throughput of 40 Mbps or less is 46.4% and 35.6% for the case or rule-based and Q-learning-based coexistence schemes solutions. However, the probability of getting this range of aggregated throughput is only 24.8% for the case of ER Q-learning and RER Q-learning-based solutions. In general, the graph shows that in terms of aggregated throughput, the ER Q-learning and RER Q-learning-based solutions outperform the other proposed alternative solutions.

Figure 5a shows the average fairness values for the rule-based, traditional Q-learning-based, ER Q-learning-based, and RER Q-learning-based solutions. The bar graph shows that the ER Q-learning-based and RER Q-learning-based coexistence schemes lead to a better coexistence in terms of the obtained average fairness. The average fairness is computed by taking an average of the fairness values obtained in each fixed load duration in the entire simulation period. The fairness value in each fixed traffic load duration is computed using Equation (Equation 3). Figure 5b shows a histogram of the fairness values obtained in each fixed traffic load duration $T_{D}$ during the simulation period. From the histogram, it can be observed that the ER Q-learning and RER Q-learning-based solutions perform better than the other coexistence solutions in terms of fairness between the technologies.

For the fairness performance evaluation, each LTE UE or WiFi station generates random data rates, which are randomly picked between {50 kbps, 500 kbps, 1 Mbps, 2 Mbps, 4 Mbps}. Likewise the throughput performance evaluation, each randomly generated data rate remains constant for $T_{D}$ seconds where $T_{D}$ is randomly picked from 5–15 s. This is done for a simulation time of 300 s and it is repeated for 100 iterations. The coexistence solutions are used to select the best actions while the offered traffic changes dynamically. The fairness performance of an action selected based on the optimal policy is evaluated by computing its fairness using Equation (Equation 5). For validation purposes, we also determine the rank of the each selected action (interms of fairness) as compared to the other possible actions. For each fixed load traffic of LTE and WiFi networks, the selected action, the obtained and target throughput, $T_{D}$, and the fairness value are stored for comparison purposes. Following this, all the other possible actions are also executed sequentially by manual selection and the fairness value is stored for every action. This is repeated for every fixed traffic generated during all the iterations of the simulation period. Finally, the rank of the action selected by the policy is determined based on its fairness as compared to the other possible actions which are selected manually.

Table 4 shows the distribution of rank of actions taken based on the traditional Q-learning, ER Q-learning, and RER Q-learning schemes in 300 s simulation with dynamic traffic. The table only presents the comparison of the reinforcement learning based solutions as the rule-based solution always leads to the best action despite the fact that it requires multiple trials until it reaches the best action.

The table shows that the traditional optimal policy of the Q-learning based solution can only get the best action at a probability of 66.7%. This low performance occurs due to the dynamic nature of the environment. The ER Q-learning and RER Q-learning based solutions on the other hand have better performance leading to the best action at a probability of 86.3% and 89.4%, respectively. Even though there is a significant number of decisions where the second-best actions are executed by the optimal policy, the performance of the coexistence schemes is not highly degraded (as the second-best action can still lead to sufficient fairness and aggregated throughput).

## 7. Conclusions and Future Work

In cellular networks, broadband applications that employ licensed spectra are expanding and the demand for high-throughput services is expected to rise. As a result, licensed spectra are becoming increasingly scarce. LTE is expected to use the unlicensed band for some of its transmissions to alleviate the spectrum scarcity problem. To effectively use the unlicensed spectrum, however, a number of issues must be addressed. The most significant issue is establishing a harmonious coexistence with the WiFi networks that already exist on the unlicensed spectrum.

Many coexistence strategies have been proposed to ensure that WiFi and LTE networks coexist together. Convolutional Neural Network (CNN) and Q-learning-based machine learning algorithms are mostly used in current coexistence methods to improve performance. Most existing coexistence systems take into account coordinated LTE and WiFi networks, in which a co-located LTE network picks its transmission time based on the quantity of WiFi traffic generated in its collision domain. The WiFi traffic demand is determined by a central coordinator which can communicate with co-located networks using a collaboration protocol to exchange status and requirements.

Collaboration protocol-based information exchange to identify traffic status requires changes to the infrastructures of the co-located WiFi and LTE networks. Moreover, adopting this inter-technology collaboration protocol increases the coexistence schemes’ complexity, traffic overhead, and reaction time. As a result, we present a coexistence scheme that can operate in uncoordinated LTE and WiFi networks without the necessity for a collaboration protocol to exchange traffic status between the technologies. In the proposed scheme, the LTE-U deploys a WiFi saturation sensing model to estimate the traffic demand of co-located WiFi networks. In particular, RER and ER Q-learning based non-cooperative coexistence schemes are proposed and evaluated. We also compare the performance of the proposed schemes with each other and with other non-coordinated schemes which use rule-based and Q-learning based approaches. The comparison is carried out in terms of model complexity, convergence, the accuracy of best action selection, fairness, and throughput. Our results show that the proposed RER Q-learning scheme converges faster than the ER Q-learning scheme and gives better aggregated throughput and fairness performance as compared to the rule-based and Q-learning schemes.

In the future, a system that integrates the traffic prediction model and the proposed coexistence scheme can be developed to enhance the overall system performance. By integrating the traffic prediction model with the proposed coexistence scheme, the optimal execution frequency of the coexistence decision can be determined. In other words, the coexistence scheme is executed every period of time *T*, where *T* is optimally selected considering computational complexity and the traffic dynamics of the co-located networks estimated by a network traffic prediction model.

## Figures and Tables

**Figure 1 sensors-21-06977-f001:**
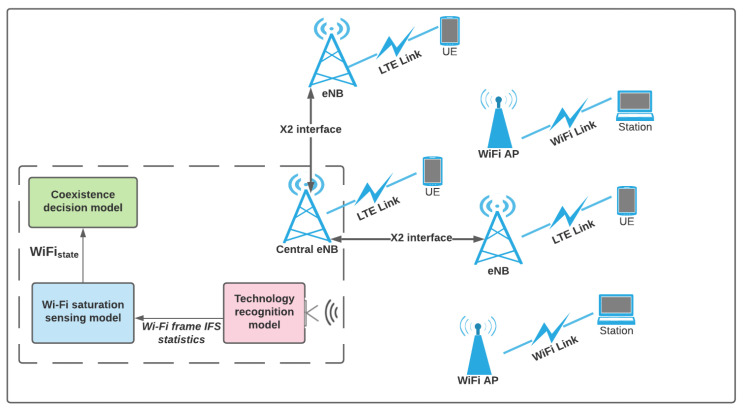
Topology of proposed Coexistence scheme for uncoordinated LTE-U and WiFi networks.

**Figure 2 sensors-21-06977-f002:**
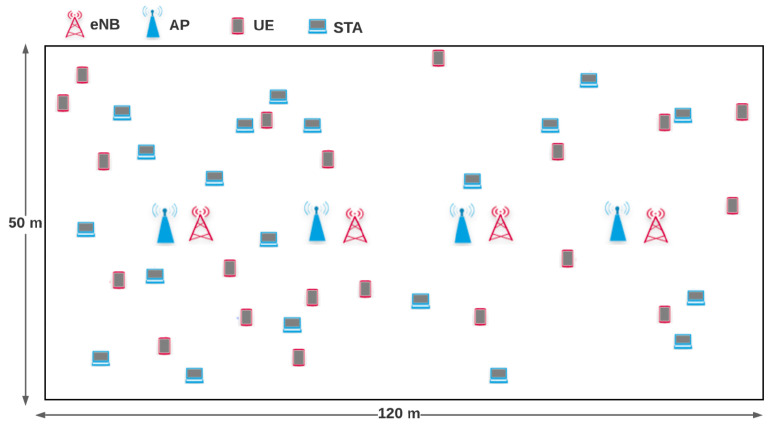
Indoor network scenario based on 3GPP specification [38].

**Figure 3 sensors-21-06977-f003:**
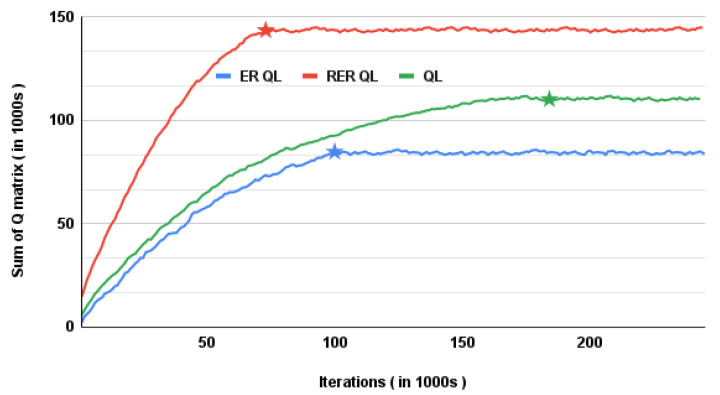
Convergence of Q-learning, ER Q-learning, and RER Q-learning based schemes.

**Figure 4 sensors-21-06977-f004:**
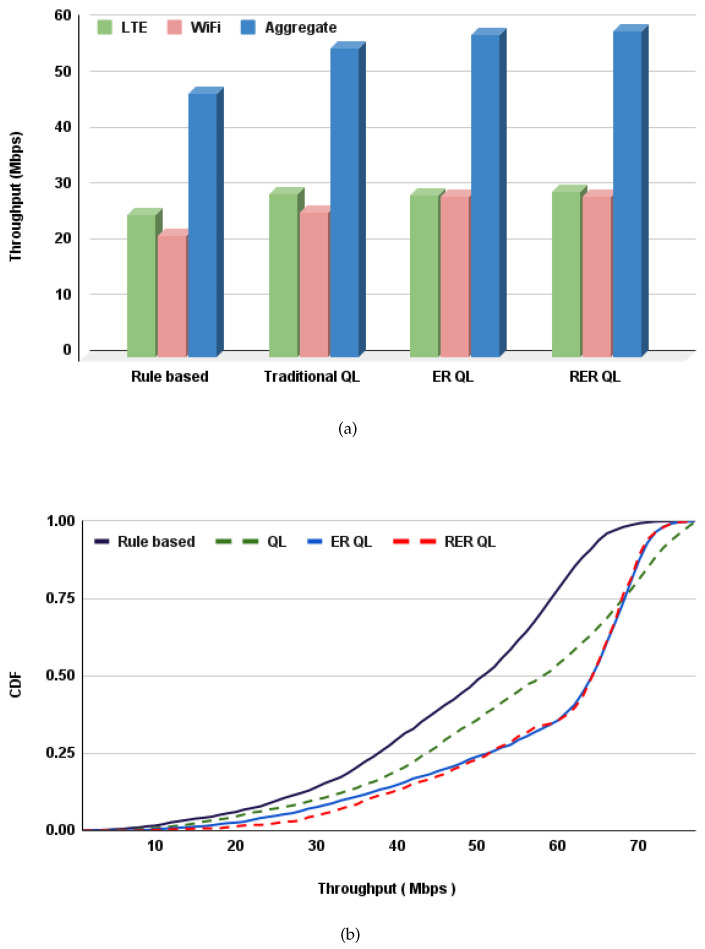
Comparison of aggregated throughput for the proposed solutions (**a**) average aggregated throughput (**b**) CDF of aggregated throughput.

**Figure 5 sensors-21-06977-f005:**
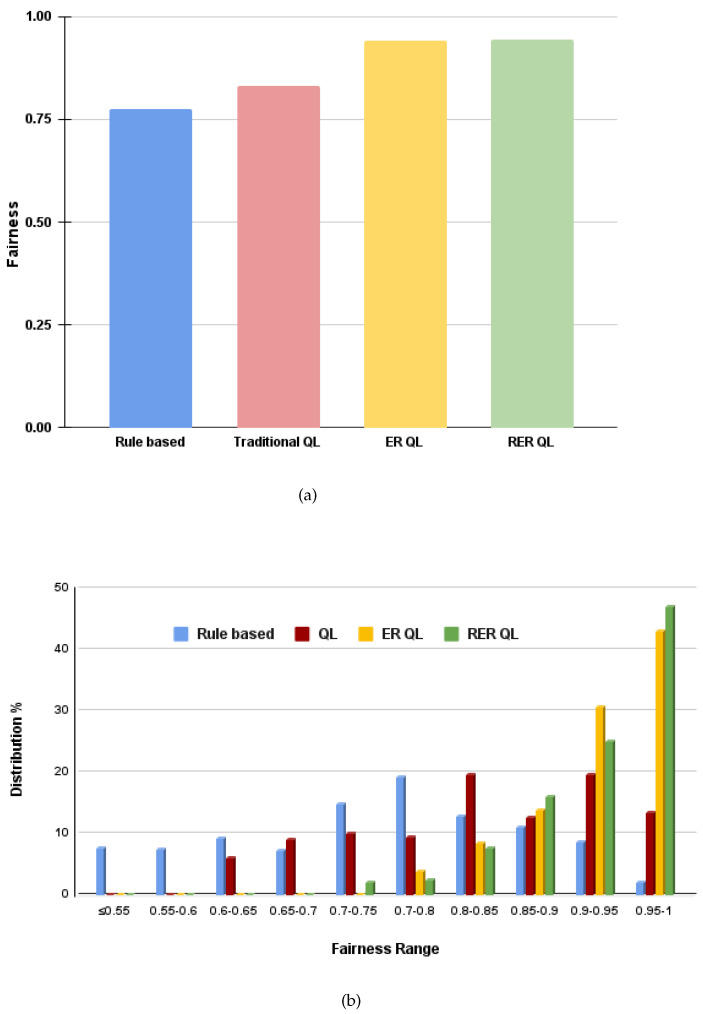
Comparison of fairness for the proposed solutions (**a**) average fairness (**b**) histogram of fairness values obtained in each fixed load duration in the entire simulation period.

**Table 1 sensors-21-06977-t001:** Summary of notations used.

Notation	Description	Notation	Description
$LTE_{\mathrm{Of}}$	LTE offered data rate	ε	Exploration/exploitation policy value
$LTE_{\mathrm{Ob}}$	LTE obtained throughput	$N_{\varepsilon }$	Number of iterations to update ε
$LTE_{\mathrm{Tar}}$	LTE target throughput	$f_{\varepsilon }$	Reduction factor of ε
$LTE_{\mathrm{Ref}}$	LTE reference throughput	$\varepsilon_{\min}$	Minimum value of ε
$LTE_{\mathrm{State}}$	LTE state	$R_{f}$	Record of experiences in a fixed traffic load
$WiFi_{\mathrm{State}}$	WiFi state	$R_{E}$	Set of record of experiences for dynamic traffic
$WiFi_{\mathrm{Of}}$	WiFi offered data rate	$N_{E}$	Buffer size of experience replay
W_S_	WiFi saturated state	$WiFi_{\mathrm{Ob}}$	WiFi obtained throughput
W_U_	WiFi unsaturated state	$WiFi_{\mathrm{Tar}}$	WiFi target throughput
γ	Discount factor	$WiFi_{\mathrm{Ref}}$	WiFi reference throughput
η	Learning rate	$N_{b}$	Number of LTE blank subframes
*S*	State space	$a,s,r,s^{\prime }$	Action, state, reward, new state
*A*	Action space	$T_{D}$	Fixed traffic load duration
$f_{f}$	Fairness factor	*K*	Number of replays

**Table 2 sensors-21-06977-t002:** LTE states and β values.

LTE State	Description	β
L_1_	*LTE*_obtained_ ≤ 10% of *LTE*_Target_	0.1
L_2_	10% of *LTE*_Target_ < *LTE*_obtained_ ≤ 20% of *LTE*_Target_	0.2
L_3_	20% of *LTE*_Target_ < *LTE*_obtained_ ≤ 30% of *LTE*_Target_	0.3
L_4_	30% of *LTE*_Target_ < *LTE*_obtained_ ≤ 40% of *LTE*_Target_	0.4
L_5_	40% of *LTE*_Target_ < *LTE*_obtained_ ≤ 50% of *LTE*_Target_	0.5
L_6_	50% of *LTE*_Target_ < *LTE*_obtained_ ≤ 60% of *LTE*_Target_	0.6
L_7_	60% of *LTE*_Target_ < *LTE*_obtained_ ≤ 70% of *LTE*_Target_	0.7
L_8_	70% of *LTE*_Target_ < *LTE*_obtained_ ≤ 80% of *LTE*_Target_	0.8
L_9_	80% of *LTE*_Target_ < *LTE*_obtained_ ≤ 90% of *LTE*_Target_	0.9
*L* _10_	*LTE*_obtained_ > 90% of *LTE*_Target_	1

**Table 3 sensors-21-06977-t003:** Common parameters used to model co-located IEEE802.11n and LTE networks [38].

Parameter	Value
Bandwidth (MHz)	20
Carrier frequency (MHz)	5180
Packet size (bytes)	1500
Traffic model	UDP
UDP rate	{50 kbps, 500 kbps,1 Mbps, 2 Mbps, 4 Mbps}
MIMO	NA
Pathloss, shadowing and fading	ITU InH
Tx Power	18 dBm
BS/AP Antenna gain	5 dBi
UE antenna gain	0 dBi

**Table 4 sensors-21-06977-t004:** Distribution of rank of actions taken based on the Q-learning, ER Q-learning, and RER Q-learning based schemes.

Coexistence	Distribution of Actions Taken in Each Rank (in %)								
Scheme	1	2	3	4	5	6	7	8	9
QL	66.7	28.1	0.5	0.2	-	-	-	-	-
ER QL	86.3	13.1	0.6	-	-	-	-	-	-
RER QL	89.4	9.7	0.9	-	-	-	-	-	-

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
