# Peer review of "Coexistence Scheme for Uncoordinated LTE and WiFi Networks Using Experience Replay Based Q-Learning"

_sensors, 2021, doi:10.3390/s21216977_

Round 1
Reviewer 1 Report
The paper presents Q-learning based techniques for improving the cooperetation of uncoordinate LTE and WiFi networks in terms of the throughput, fairness and computational complexity. The research topic under investigatio is timely and despite the fact that many papers in this area exist, including several ones of the authors themselves, the paper offers a satisfactory contribution. Moreover it is well written.
Author Response
-
Reviewer 2 Report
The authors have proposed two techniques, namely, Experience Replay (ER) and Reward selective Experience Replay (RER) based Q-learning as solution for
the coexistence of uncoordinated LTE-U and WiFi networks. Only a few comments and suggestions.
In the Abstract, rephrase the last paragraph, startng with "The simulation results ..." because it is not clear.
pp. 600 89.4% and 86.3% respectively -> 86.3% and 89.4% respectively
In the Conclusions, eliminate the last paragraph or indicated that it will be your Future Work.
Author Response
We appreciate the reviewer's suggestions and we have modified the errors and rephrased the last paragraph of the conclusion, in such a way that it clearly states future work directions. The following modified future work description is included in Section 7 (page 21):
“In the future, a system that integrates the traffic prediction model and the proposed coexistence scheme can be developed to enhance the system performance. By integrating the traffic prediction model with the proposed coexistence scheme, the optimal execution frequency of the coexistence decision can be determined. In other words, the coexistence scheme is executed every period T, where T is optimally selected considering computational complexity and the traffic dynamics of the co-located networks estimated by a network traffic prediction model.”
Reviewer 3 Report
The paper presents the coexistence scheme for uncoordinated LTE and WiFi
networks using experience replay based Q-learning. The paper is interesting, however some matters need to be addressed before publication.
- A better explanation of your contribution needs to be added in the Introduction Section. Please set differences of your work with respect to previous work. Please highligh the advantages, benefits and contributions of the new tecnique and results.
- Provide more details of the methodology used.
- Although the authors present simualtion conditions, a more datailed explanation is needed for system model and simulations. Please set difference between the different techniques illustrated.
- The performance evaluation is achieved by throughput metrics. Can you highlight why to use this kind of metrics? The difference with respect to others.
- It is needed a better description of the results. Please highligh the benefits, the advantages and contributions of your proposal.
